# Pulse Consumption among Portuguese Adults: Potential Drivers and Barriers towards a Sustainable Diet

**DOI:** 10.3390/nu12113336

**Published:** 2020-10-30

**Authors:** Mariana Duarte, Marta Vasconcelos, Elisabete Pinto

**Affiliations:** 1CBQF—Centro de Biotecnologia e Química Fina—Laboratório Associado, Escola Superior de Biotecnologia, Universidade Católica Portuguesa, 4169-005 Porto, Portugal; marianacrduarte@hotmail.com (M.D.); mvasconcelos@porto.ucp.pt (M.V.); 2EPIUnit—Instituto de Saúde Pública, Universidade do Porto, 4050-091 Porto, Portugal

**Keywords:** environmental sustainability, pulses, protein substitutes

## Abstract

The transition from diets rich in animal products to plant-based protein foods—like pulses—is crucial, for both environmental sustainability and human health. The aim of this study was to characterize the current consumption and to describe the drivers of and barriers to pulse intake in Portugal. Using a quantitative approach, a semi-structured questionnaire was distributed online, and 1174 valid responses were obtained. The most consumed pulses were beans and peas, consumed at least once a week by 48.3% and 44.4% of the sample, respectively. When participants were asked about the possibility of replacement, even partially, of animal products for pulses, 15.0% stated they would not substitute even in a food scarcity scenario. In the qualitative study, ten individuals involved at different steps of pulses’ supply and value chain were interviewed in order to study individual behaviors and experiences linked knowledge and consumption of pulses. It was noticed that the lack of recognition of their nutritional value, the high cooking time and the effect of the anti-nutritional factors were commonly pointed out as barriers. The identification and understanding of perceived barriers for that low consumption will leverage the development of new strategies to promote this promising alternative.

## 1. Introduction

World population growth poses new questions as it implies an increase in food production in a lower amount of arable land. According to the United Nations Department of Economic and Social Affairs, the world population reached 7.7 billion in 2019 [1]. In line with current forecasts, the world population will hit 8.5 billion in 2030, 9.7 billion in 2050, and in 2100, the value will be closer to 10.9 billion [1,2,3].

This population growth brings new challenges and opportunities for a demographically modified society [3,4]. Increased food availability will be necessary in order to ensure the nutritional requirements for millions of people in different stages of their life cycle [3,5]. Therefore, it will be necessary to rethink the food pattern that is followed today by almost all modern societies, including the majority of the Portuguese population.

In 2017, the Portuguese Food Balance Sheet (PFBS) disclosed that “Meat, fish and eggs” food group had the most significant deviations from the Food Wheel (the Portuguese food guide), with an availability of 11.5% above the recommended consumption. By contrast, the pulses group had a deficient availability, accounting for only 0.6% of the total amount of recommended foods, when the recommendation is 4% [6]. This low consumption may be a result of the current low adherence to the Mediterranean diet, an ancestral and healthy food pattern practiced by most of the Portuguese population for centuries and that is characterized by a substantial intake of pulses [7]. Notwithstanding the scarcity of national data on food consumption throughout the time, data on food availability (food balance sheets) shows clearly a decrease in pulses availability [8].

The drivers which lead to this meat consumption pattern cannot remain detached from the ecological footprint associated with livestock production [9,10,11]. According to Sabate and collaborators, to produce 1 kg of bovine protein, ten times more water is necessary than the amount needed to produce the same quantity of vegetable protein, like the one from pulses [12]. In this way, the development of efficient strategies that allow satisfaction of nutritional needs, without damaging the environment, is crucial [8,9]. An example of a “green” attitude towards our planet will be the adoption of sustainable diets-defined by the Food and Agriculture Organization of the United Nations (FAO) as “diets with low environmental impacts which contribute to food and nutrition security and to healthy life for present and future generations. Sustainable diets are protective and respectful of biodiversity and ecosystems, culturally acceptable, accessible, economically fair and affordable; nutritionally adequate, safe and healthy; while optimizing natural and human resources” [4,13,14].

Pulses are a food group with an important role in human nutrition [15]. These nutritious seeds are a significant source of vegetable protein, have good carbohydrates (rich in starch) and mineral content and have a low-fat content [14,16]. Although pulse protein has a lower biological value than animal protein, it can be perfectly complemented with cereal protein—this match allows to reach a good supply of every essential amino acid [15,16,17]. Besides protein, among the micronutrients present in pulses, the B complex vitamins and minerals such as calcium, iron, potassium, magnesium and zinc stand out [15,18]. These foods have also other bioactive substances like phenolic compounds, isoflavones and other antioxidants, with interesting metabolic effects [15,19].

Despite the advantages mentioned above, in most parts of the world, pulses are greatly undervalued and poorly recognized as a substitute for animal protein [15]. Having this in consideration, many researchers have been looking at potential barriers to pulse consumption [10,20,21]. However, some of these barriers are cultural and should be studied in each population in order to be overcome.

To date, there are few studies evaluating the real predisposition to the inclusion of pulses as protein substitutes, even if partial, for meat or fish products, namely in Portugal. The aim of this study was to characterize current pulse consumption in a sample of adult Portuguese population and to describe the potential drivers and barriers to the inclusion of this source of protein in the Portuguese diet.

## 2. Materials and Methods

### 2.1. Quantitative Data Collection

In this research, quantitative and qualitative data were collected. Firstly, a self-administered questionnaire, with both closed and open-ended questions, entitled “New Protein Sources in Portuguese Diet”, was disclosed through e-mail and Facebook, using the snowball sampling method [22], over a period of ten days. The questionnaire was distributed amongst a Faculty community (~800 individuals, including students, academic and administrative staff, and researchers) and subsequently among social networks of main researchers. This contact had two objectives: to request participation in the study and to request its widespread distribution among participants’ contacts. The inclusion criteria for participation were people living in Portugal aged at least 18 years old. The introduction to the questionnaire mentioned these inclusion criteria. Later, in the questions about sociodemographic data, the respondent’s age and geographic location were asked. In case of not meeting the inclusion criteria, questionnaires were eliminated from the database.

The questionnaire was divided into two sections: firstly, sociodemographic data were collected (age, sex, education and local of residence); secondly, the pulses’ intake frequency was ascertained, considering six different pulses and seven options for frequency of consumption (from “never” to “more than once a day”) and a question about the readiness to include pulses as a partial substitute of animal protein, through a closed question with five options of answer. The self-perception of food pattern was also ascertained. The questionnaire was prepared in Google forms, and the respective link was distributed. The average filling time was five minutes. A total of 1174 valid responses were obtained.

### 2.2. Qualitative Data Collection

Using a qualitative approach, in order to address more information and have a deep description about the drivers and barriers for the pulses’ consumption, a group of ten individuals, selected according to their professional intervention’ areas, were interviewed. Once we intended to interview people dealing with pulses at different steps of the value chain, from production to the consumption, first, we identified the stage that we want to be represented, and second, we identified a person working in this specific stage. This heterogeneous group was composed by representative elements of this sustainable food chain, focusing on pulses. The respondents were linked to teaching and research works, to industry, to agriculture, to the Food and Agriculture Organization (FAO), to a nutritionists’ professional association and to a multinational company that deals with pulse commercialization. There were also two consumers—one professionally active and one without professional activity (“housewife”). Using these face-to-face interviews, data was collected on individuals’ behaviors, attitudes and experiences particularly linked to pulses’ consumption. All participants answered the same set of six open-ended questions (Table 1). Interviews were recorded and transcribed for the principal researcher. Afterwards, the audio records were destroyed. All participants signed an informed consent. The principles of the consolidated criteria for reporting qualitative research (COREQ) were assured in this research [23]. Our study was approved by the Institute of Bioethics, Catholic University of Portugal, through the Ethics Screening Report 11/2017.

### 2.3. Statistical Analyses

SPSS version 24 (IBM Corp. Released 2016. IBM SPSS Statistics for Windows, Version 24.0. Armonk, NY: IBM Corp.) was used for the statistical analyses of quantitative data. Categorical variables were compared using chi-square and Fisher’s exact tests. Statistical significance was defined by a *p* < 0.05. To analyze the data of the qualitative research, the content analysis method was used [24]. We started with a pre-analysis that consist in a free-floating reading and in the elaboration of the analysis structure (categories and sub-categories), which we had defined previously. Afterwards, we explored the interviews and codified the pieces of text that can be allocated to each sub-category. Finally, we interpreted data merging theoretical concepts and empirical data. We added this procedure to the manuscript.

## 3. Results

### 3.1. Pulses’ Intake among Portuguese People—A Quantitative Approach

Respondents were mostly female (71.4%) and had a mean age of 27.2 ± 9.9 years. The majority of participants (62.0%) attended university education or had higher education. Individuals living in the North of Portugal were overrepresented (78.5%). Moreover, only 3.8% of respondents followed a vegetarian dietary pattern. Descriptive statistics of the sample are presented in Table 2.

The most consumed pulses were beans and peas, consumed at least once a week by 48.3% and 44.4% of the sample, respectively. Soybean and fava bean were much less consumed, with proportions of non-consumption of 48.5% and 46.6%, respectively. Finally, 58.9% of the respondents reported “never” consuming lentils in their daily diet. There were no statistically significant differences in the frequencies of pulses’ consumption between sexes. These results are presented in Table 3.

To the question “Have you ever considered the possibility of replacing meat or fish by pulses in some meals?”, 15.0% stated they would not substitute, even in a food scarcity scenario, 42.0% answered that it would be easy for them to do so and 20.0% already did (Figure 1).

There were nearly twice more women who reported using pulses as a meat and fish substitutes, compared to men (22.8% vs. 12.7%). Moreover, there was a growing gradient of reluctance to change eating habits among males [“I already do it”: 12.7% vs. “I do not consider this possibility”: 25.3%] (Table 4).

In addition, it was observed that the predisposition to include pulses as protein substitutes for meat and fish increased with age, corresponding to 79.4%, 86.3% and 90.6% of participants aged up to 21 years, from 22 to 27 years and 28 years or older, respectively (*p* < 0.001) (Table 5). When evaluating the influence of education level on the possibility of this substitution, it was found that among those who already reported making this food change, the most educated people (≥13 years of schooling vs. a lower education) were the predominant group (23.4% vs. 14.2%) (Table 3).

Non-omnivores respondents reported much more frequently than omnivores that they already used pulses to replace meat or fish (83.6% vs. 17.4%) (Table 4).

There were statistically significant differences between gender regarding the desire for the inclusion of pulses in their diet (89.1% of the women and 74.7% of the men, *p* < 0.001) (Table 5).

### 3.2. Knowledge and Perceptions Regarding Pulses: Qualitative Study

The first two questions of the qualitative study intended to evaluate the knowledge about the diversity of pulses’ species and varieties and to understand the frequency of their consumption. All respondents remembered beans as a human food pulse (*n* = 10). Moreover, chickpeas and peas have been frequently reported (*n* = 8). On the other hand, lupine and grass pea were the less mentioned pulses grains, listed only by four and three participants, respectively. Recognized for their oil content, the soybean and peanut are poorly recognized as food belonging to the pulses’ group.

As observed in the quantitative approach, the pulses with higher intake prevalence were beans and then chickpeas. On the other hand, although fava bean had been remembered as a pulse, no one said it was eaten on a daily basis. Only one person reported eating lentils, and nobody mentioned the consumption of soybean and peanut.

Among those who showed greater knowledge about the pulses’ species were the participants linked to teaching and research and the respondent linked to an agriculture practice.

In the second part of the interview, an image of several pulses grains was showed in order to ask what the sensations/feelings it aroused were, or which experiences and memories were associated with this food’s consumption (Figure 2).

Half of the respondents reported associating pulses with traditional dishes—one of the citations, said by an expert in this research area, was “essentially it is an image that relates to our food tradition, and this is very important”. Another interviewee, the housewife, said that the image “evokes some good sensations such as peas, peas and eggs, which were made at my parents’ house”.

Pulses were also recorded with nostalgia, linked with memories of ancient times (“it reminds me of childhood, traditional flavors, grandma’s house”) or feelings of pleasure and well-being (“I always connect this foodstuff with cold and comfort”). Although less mentioned, it is relevant to note that two people had pointed out pulse grains as an important element for a well-balanced diet and said that “beauty, color, diversity, balance, variety…” were key characteristics of this food. The person who had a job in management of the wholesale and retail trade of a multinational enterprise said, “the image awakes to a more conscious food pattern”.

According to the Portuguese Food Wheel, the consumption of pulses must be one or two servings per day. One portion represents one tablespoon of raw dried pulses (~25 g) or three tablespoons of dried or fresh pulses, after cooking (~80 g).

Thus, when the knowledge about this consumption’s requirements was analyzed, it was possible to understand that the interviewees who were aware of these recommendations were industry-related people (linked to new product development), the teacher of higher education, the corporate representative of a public professional association and a nutritionist in a collective food service establishment. On the other hand, six of the ten participants interviewed showed not to know the exact recommendations, including three of them who revealed a total lack of knowledge (“… I have no idea what is the daily value”).

Regarding the compliance with these recommendations, it was found that only two persons mentioned they were ingesting this food as recommended. Among the reasons pointed out for that low pulse intake were the lack of knowledge about the great variety of cooking recipes with this ingredient—one respondent said “my food recipes not always match with pulses”, another reported “usually I put it in the soup (…) but not always”.

The FAO representative said “I forget that pulses exist”. The second most pointed reason was the preparation’ limitations—the element who works in a food service establishment claimed “we need to think a little when we want to make pulses… I don’t like canned beans… we need to plan meals with pulses and soak them previously”. The interviewed researcher said, “there were social issues linked to a low consumption of pulses; the speed in our daily life does not allow us to use pulses more often”. It should also be noted that one person, linked to the product’s development, did not eat pulses due to an intolerance or other chronic digestion problems—“I feel bad when I eat beans… when I try to eat them I have gastrointestinal problems”.

Finally, when the respondents were asked whether they knew the explanation for current pulses’ consumption recommendations, they all answered affirmatively. Among the reasons listed for this counseling were “nutritional quality” (*n* = 8), the presence in pulses of a “good source of protein” (*n* = 7) and the concern with environmental sustainability—pulses were considered to be “a good substitute for meat and other animal products...” (*n* = 3).

In the last part of the interview, the participants were asked to give their opinion about the reasons that justify the low consumption of pulses in Portugal. This was the question where the respondents had longer pauses in their speech and showed more doubts. The main reason that was pointed out was the lack of nutritional literacy by the Portuguese population, as the following sentences show: “lack of knowledge of their value”/“Portuguese people are not well informed”/“maybe either because we don’t recognize their importance (…)”. Moreover, the increase in purchasing power was also mentioned. One person stated “(…) it’s a foodstuff which lost their reputation… Not many years ago there were serious economic crises affecting food availability”; another element said “in the past, maybe in our grandparent’s time, people did not have money to eat meat and fish in their daily life or the portions were lower. So, they ate more pulses, there were pulses in all homes, at least, a chickpea soup…”.

The food myths were also mentioned as a barrier by half of the interviewed people, as illustrated in the following sentences “pulses are linked to a poor’s man food…”; “ (…) many times we connect the word pulse to a heavy food (…) causing some gastrointestinal disorders (…).” Four respondents, also, said that food preparation was another problem of this foodstuff because “people do not have time…” and “pulse grains require a huge processing time”.

In order to promote pulses’ intake, the majority of respondents (*n* = 8) referred that it was necessary to spread information about their benefits—“more media disclosure…”; “… [need] a certain advertising campaign…”. Moreover, four participants mentioned that it was important to invest in food education, alerting people to the need to raise “awareness among health professionals…” and “educate the general population”. Among the given opinions, the importance of demystifying wrong ideas linked with pulses’ intake was emphasized—“we need to clarify some concepts and put out myths, currently expressed by common sense”.

Two respondents talked about the urge of making food policies that instigate pulse grains’ production and their consumption, evident in the following quotes: “possibly some legislation… it is an issue of public policy…” and “…it was necessary to reduce the use of salt in food industry [canned pulses]”.

Finally, it was possible to verify that one of the most suggested practical ideas (*n* = 5) was the need to develop new culinary recipes (“we have to give different formulations”; “a cookbook with the intervention of some famous chefs…”). This section may be divided by subheadings. It should provide a concise and precise description of the experimental results, their interpretation as well as the experimental conclusions that can be drawn.

## 4. Discussion

According to the World Cancer Research Fund (WCRF), between 1961 and 2013, the amount of meat available for consumption per capita doubled, from 23 to 43 kg/year [25]. In Europe, in 2013, the amount of meat available for consumption was 77 kg/per capita/year, with an average protein availability of 140g/per capita/day in Portugal [25,26]. Thus, having in consideration economic development was an important driving force for increasing meat intake and cattle’s enteric fermentation is the most representative portion for carbon dioxide emission, it is crucial to look for alternatives that are healthy as well as sustainable—such as pulses, algae and insects [25,26,27].

Due to its geography, Mediterranean diet was practiced in Portugal, and it is characterized by a set of healthy dietary habits that include high consumption of unprocessed and plant foods, such as grains, nuts, pulses, fresh and seasonal fruits and olive oil as the main source of fat. In addition, Mediterranean diet is characterized by its frugality and moderation, and it also stimulates conviviality during meals [28]. However, in the last years, as a consequence of economic development and globalization, we assisted to a nutrition transition, with a decrease in the consumption of plant foods and an increase in animal foods and processed foods, as well as a loss of the characteristics of frugality, moderation and preference for seasonal foods [29]. The decrease in pulses’ consumption is experienced by other countries, like France [30].

Regarding the consumption of pulses, in this project, it was possible to see that, as it would be expected, the pulse most commonly eaten in Portugal is the common bean (*Phaseolus vulgaris* L.), curiously as it happens in a study conducted in Poland [31]. According to the scientific literature, this is the most consumed species worldwide, representing 75% of the total of dried pulses consumed in Portugal [27,32,33]. The popularity of common beans may be justified by the greater familiarity with this pulse, which appears as a main ingredient in several traditional Portuguese dishes (such as “Tripas à Moda do Porto” (cow bowel with white beans, typically consumed in Porto, North of Portugal) and “Feijoada à Transmontana” (several types of meat, including pork, sausages, chicken, vegetables and red beans, typically consumed in Trás-os-Montes, a region in North of Portugal)). The popularity of beans was enhanced in other studies conducted in France or Australia [21,31].

When studying the receptivity of the Portuguese to the replacement of animal protein sources by pulses, it was noticed that, although in our days it is not a common practice among omnivorous individuals, a large percentage declare this exchange would be easy to achieve or even they already do it. The same willingness was observed in a Polish study [34].

Analyzing the influence of sociodemographic factors in the predisposition to include pulses as protein alternatives to the consumption of meat or fish, it was found that both gender and educational level are important elements to have in mind. The statistically significant difference found between men and women allowed us to realize that women are more likely to perform this modification of eating behavior [35,36]. Men are more reluctant probably because, for them, meat products represent masculinity [37]. Meat consumption is associated with physical strength and power, symbolic values very often attributed to males [36,37,38]. As for the importance of literacy, it was possible to see that respondents whose education level was higher than the 12th grade were more predisposed to increase the consumption of pulses instead of eating animal products. Generally, people with more years of schooling tend to have a higher quality diet, to seek information on healthier alternatives and to invest more in their health [39].

Given this predisposition to change eating behavior, we tried to understand what the reasons are for the low consumption of this food group compared to the recommendations stated in the Food Wheel [17,33,40]. Among the potential barriers identified during the interviews conducted, the lack of food and nutritional literacy of the consumer stood out. The lack of knowledge of the nutritional value and different consumption alternatives and the current food myths have kept pulses away from the Portuguese households. Several of these limitations were also registered in previous studies [21,30].

According to what was stated, for pulses to stop being understood as a “food for the poor” or the “meat of the poor”, it is important to develop strategies for the promotion of their consumption, following the example of the FAO initiative, which defined 2016 as the International Year of Pulses [15,41]. Another example is scientific consortiums that tried to enforce the role of pulses in food chains, like the European project with the acronym TRUE—Transition Paths to Sustainable Legume-based Systems in Europe—which has been looking for new ways to increase the growth and consumption of pulses [40]. In another example, a catering company in Portugal implemented a project called “Choose Beans” aiming to encourage the intake of pulses, through an increase in their availability, the development of workshops and also the creation of teaching materials [40,42]. This strategy was successful in increasing the pulse intake in the units that received this campaign by 25% [40].

With the current work, it is possible to acknowledge the need to develop or reinvent existing recipes so that pulses become more appreciated and less associated with “heavy” dishes [43]. In this perspective, some chefs, who understand the need to change consumption patterns, have shared their culinary secrets to make pulses more appealing and, therefore, encourage people to increase their intake [15]. Some of them started by sharing recipes that had meat products as the main ingredient, turning them into vegetarian, more sustainable options [42,43,44]—such as the creation of recipes for chickpea meatballs, red bean croquettes and lentil hamburgers [42].

Considering other false concepts commonly pointed out as barriers to the consumption of pulses, it is important to highlight that, during the course of the interviews, one of the most enumerated obstacles was the fact that these foods are not in line with the current lifestyle in modern societies, where the concepts “fast” and “practical” are increasingly common [21,43,44,45]. There are still many consumers who are not familiar with the option to buy canned pulses instead of dried pulses, sold packaged or in bulk [46]. This creates a huge obstacle to their use since it requires increased preparation and cooking [45,46,47]. However, in a study conducted in Poland, being able to buy easy to prepare pulses was pointed out as a key reason to eat pulses [31], while a study conducted in France corroborated the difficulty in pulses’ cooking [30], enhancing the role of culture and lifestyle in this appreciation.

Some of the popular beliefs pointed to the consumption of canned products are the possibility of migration of components from the (metallic) packaging to the food, the presence of microorganisms, the loss of nutritional value and the high content of sugar and salt [45,48]. From the nutritional point of view, since the pulses are subjected to a heat treatment inside the packaging itself, the loss of nutrients is not significant [45,48]. Perhaps the biggest concern is, in fact, the high content of sodium chloride (salt) in canned foods [45,46]. Considering this, although the reduction of added salt represents a challenge for the food industry from a technological and sensory point of view, it is essential to develop and implement food policies that obligate to reformulate the composition of these food products [45]. In addition, it is important to develop education campaigns for the population in order to provide efficient strategies to minimize this unreasonable salt consumption—for example, teaching how to read food labels and advising draining the liquid from the can have been pointed out in different scientific works [15,46].

Nevertheless, according to several publications on this theme, canned products are a safe and economical alternative, so regardless of the form of consumption, pulses should be incorporated more frequently in the diet [30,46,48,49].

In addition to what was mentioned above, another drawback pointed out by the respondents was related to the gastrointestinal effects of pulses, such as flatulence, abdominal discomfort, cramps and/or diarrhea [15,19,49,50]. These symptoms are often associated with the inability of the human being to break the α-glycosidic bonds that exist in oligosaccharides—raffinose, stachyose and verbascose, for example—which are part of the nutritional composition of pulses. Thus, strategies to mitigate these digestive troubles have been sought and identified [47,50,51]. Among the possible solutions for the eliminating these non-nutrients, the process of soaking and cooking pulses stands out, since, in addition to improving the palatability of these foods, they reduce the presence of thermolabile components—such as lectins and inhibitors of trypsin—and the content in the above-mentioned oligosaccharides [19,49,50,51].

Lastly, notwithstanding the valuable inputs that this paper adds, it is important to identify some of its limitations. The present study helps to fill gaps in scientific literature on the motivations that lead to the current drop in pulse consumption. It provides answers to questions regarding prevalence of willingness for the replacement of meat or fish by pulses and data about barriers to the consumption of this food. This useful information helps international and local companies to develop and implement new ways of looking at pulses and reinventing forms that suit the lifestyle followed by the Portuguese consumers. Moreover, this study generates new hypotheses and prompts further, tailor-made research for the Portuguese population. However, we can also highlight some limitations. The first and perhaps the most evident one concerns the selected sampling model—“snowball” sampling—since, when the participants themselves indicate other participants, they probably share certain characteristics under study, which does not guarantee the representativeness, and it makes it difficult to extrapolate the results for the general population. Moreover, in this context, the age of the respondents was another limitation since the questionnaire was filled online and disseminated through social networks or via email. This means that the percentage of older individuals was low. Another disadvantage of using this sampling technique was related to the demographic scope: By not taking a sample at random, it was noticed that the majority of respondents belonged to the region of northern Portugal, the area where the study was developed. We can speculate that if we included in the sample older people, especially men, we would obtain similar frequencies of consumption, considering the results of the national food survey conducted in 2015 [52] but probably a higher reluctance to substitute animal sources of protein by pulses. It is also important to consider that some respondents may not be able to adequately describe their usual intake of pulses. This situation probably led to an information bias.

## 5. Conclusions

From the analysis of the obtained data, it is possible to conclude that the Portuguese population may not be very receptive to the current possibility and need of replacing meat and fish with “proteins of the future”.

After identifying the main barriers to the consumption of pulses, it was realized that it is probably necessary to create food education sessions focused on nutritional, health and sustainability benefits in order to demystify concepts rooted in the Portuguese population. However, it is also fundamental to provide recipes that are not only easy and quick to prepare but also allow their inclusion in innovative and different forms of presentation from those already known and traditional.

In conclusion, for the successful replacement, even if partially, of meat by other vegetal protein alternatives, like pulses, it is essential to involve different elements representing multiple steps in the agri-food chain. Agricultural and nutritional policies should be developed or adjusted aiming to optimize food availability and the health of the population as well as reducing the harmful effects on the environment. From a practical point of view, for example, it will be necessary to readjust the rules established in order to encourage young farmers to produce pulses in a sustainable way, which means investing in polycultures and more sustainable farming methods that promote agrobiodiversity. From a nutritional standpoint, it is important to define policies that reduce the use of salt in food industry (a problem commonly highlighted by people who only choose fresh pulses—a less practical version.

## Figures and Tables

**Figure 1 nutrients-12-03336-f001:**
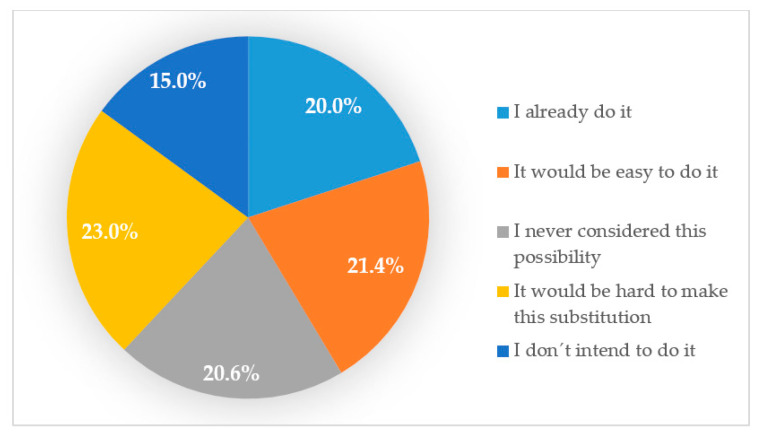
Readiness for replacement of animal products with pulses.

**Figure 2 nutrients-12-03336-f002:**
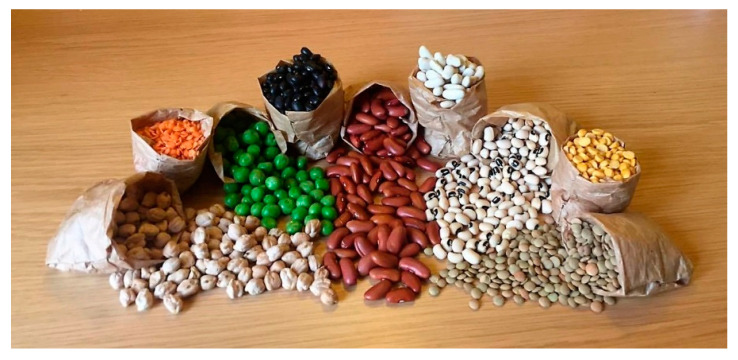
Photography used during interviews.

**Table 1 nutrients-12-03336-t001:** Questions used in the personal interviews.

Categories	Sub-Categories	Questions
Pulses	Knowledge about pulses’ variety	What pulses do you know?
	Consumption	Of the pulses you mentioned, which ones do you consume regularly, for example more than once a month?
	Feelings and Emotions	When you see this image, what sensations/feelings does it arouse? What experiences and memories do they evoke?
	Knowledge and Compliance with recommendations	Do you know if there are any recommendations for daily consumption of pulses? Are you complying with these recommendations? If not, why not?
Low Consumption of Pulses in Portugal	Explanations	In your opinion, what are the concrete reasons that justify the low consumption of pulses in Portugal?
	Strategies to Reverse the Situation	What do you think could be done to increase the consumption of pulses in Portugal?

**Table 2 nutrients-12-03336-t002:** Participants’ characteristics.

	*n* (%)
Gender	
Female	1243 (71.4)
Age (years)	
≤21	681 (39.1)
22–27	504 (28.9)
≥28	556 (31.9)
Education Level	
≤12 years	662 (38.0)
≥13 years	1079 (62.0)
Eating Pattern	
Omnivores	1674 (96.2)
Other	67 (3.8)

**Table 3 nutrients-12-03336-t003:** Consumption frequency of common pulses in Portuguese adults.

	Food Consumption Frequency			
Pulses	Never (n (%))	Less than Once a Week (n (%))	Once a Week (n (%))	More than Once a Week (n (%))
Bean	131 (7.5)	769 (44.2)	429 (24.6)	412 (23.7)
Chickpea	369 (21.2)	986 (56.6)	258 (14.8)	128 (7.4)
Lentils	1026 (58.9)	539 (31.0)	105 (6.0)	71 (4.1)
Pea	216 (12.4)	752 (43.2)	404 (23.2)	369 (21.2)
Fava bean	812 (46.6)	783 (45.0)	90 (5.2)	56 (3.2)
Soybean	845 (48.5)	602 (34.6)	122 (7.0)	172 (9.9)

**Table 4 nutrients-12-03336-t004:** Willingness for the replacement of meat or fish by pulses.

	Have You Ever Considered the Possibility of Meat or Fish Replacement by Pulses in Some Meals?				
	*“I already Do It”*	*“It Would Be Easy to Do It”*	*“I never Considered This Possibility”*	*“It Would Be Hard to Make This Substitution”*	*“I Don’t Intended to Do It”*
	*n* (%)				
Gender					
Female	284 (22.8)	294 (23.7)	267 (21.5)	263 (21.2)	135 (10.9)
Male	63 (12.7)	79 (15.9)	92 (18.5)	138 (27.7)	126 (25.3)
Age					
≤21 years	94 (13.8)	107 (15.7)	133 (19.5)	207 (30.4)	140 (20.6)
22–27 years	104 (20.6)	124 (24.6)	89 (17.7)	118 (23.4)	69 (13.7)
≥28 years	149 (26.8)	142 (25.5)	137 (24.6)	76 (13.7)	52 (9.4)
Education Level					
≤12 years	94 (14.2)	109 (16.5)	173 (26.1)	167 (25.2)	119 (18.0)
≥13 years	253 (23.4)	264 (24.5)	186 (17.2)	234 (21.7)	142 (13.2)
Food Pattern					
Omnivore	291 (17.4)	369 (22.0)	356 (21.3)	399 (23.8)	259 (15.5)
Non-Omnivore	56 (83.6)	4 (6.0)	3 (4.5)	2 (3.0)	2 (3.0)

**Table 5 nutrients-12-03336-t005:** Relation between sociodemographic characteristics and motivation to replace animal products with pulses.

	*n* (%)	*n* (%)	*p*
Gender			
Female	1108 (89.1)	135 (10.9)	<0.001
Male	372 (74.7)	126 (25.3)	
Age			
≤21 years	541 (79.4)	140 (20.6)	<0.001
22–27 years	435 (86.3)	69 (13.7)	
≥28 years	504 (90.6)	52 (9.4)	
Education Level			
≤12 years	543 (82.0)	119 (18.0)	<0.010
≥13 years	937 (86.8)	142 (13.2)	
Food Pattern			
Omnivore	1415 (84.5)	259 (15.5)	0.005
Non-Omnivore	65 (97.0)	2 (3.0)

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
