# Peer review of "Pulse Consumption among Portuguese Adults: Potential Drivers and Barriers towards a Sustainable Diet"

_nutrients, 2020, doi:10.3390/nu12113336_

Round 1
Reviewer 1 Report
The paper is very easy for the reader to read, although it has many gaps.
Authors must choose PULSE or LEGUME.
In the article, there is one nomenclature the next time another - it should be standardized.
The authors mention "hypotheses" several times in the text.
Please include them in the section describing the research material: H1, H2, ....
Just like the authors presented the profile of the respondent, please present the expert's profile in tabular form.
In the DISCUSSION section, I feel a lack of knowledge, as well as comparisons to the specific results of other researchers.
The authors can refer to research conducted in Europe on the legume market of authors such as: Śmiglak-Krajewska, Jerzak, Melendrez-Ruiz, Halicka and others, or from other countries distant from Portugal, such as Australia - Figueira.
The works of these authors are available in the MDPI or Scholar database.
From the discussion, the LIMITATION sections should be separated as a separate item.
Author Response
Response to reviewer comments
We thank all the valuable comments and the opportunity to improve our manuscript. Please, find below our answer to your comments and the respective indications about the changes operated in the manuscript. We have put the reviewers’ comments with italic font and our answer start with “A:”.
Reviewer #1: Comments for authors
The paper is very easy for the reader to read, although it has many gaps.
Authors must choose PULSE or LEGUME.
In the article, there is one nomenclature the next time another - it should be standardized.
A: We thank the reviewer for the positive comment, and we agree that the terminology should be consistent throughout. Alterations have been performed according to the suggestion. We chose “pulse” as the only term. All changes are marked with track changes in the manuscript.
The authors mention "hypotheses" several times in the text. Please include them in the section describing the research material: H1, H2,.... Just like the authors presented the profile of the respondent, please present the expert's profile in tabular form.
A: Reading again the manuscript we agree with the reviewer. Thank you for the observation. Now, this term was replaced by “to consider the possibility”. In the first version, we used “hypothesis” as a literal translation of the questions of the questionnaire.
In the DISCUSSION section, I feel a lack of knowledge, as well as comparisons to the specific results of other researchers. The authors can refer to research conducted in Europe on the legume market of authors such as: Śmiglak-Krajewska, Jerzak, Melendrez-Ruiz, Halicka and others, or from other countries distant from Portugal, such as Australia - Figueira. The works of these authors are available in the MDPI or Scholar database.
A: We thanks a lot your comment and your help. We tried to realize a deeper literature review about this topic, and we identified several additional papers that we considered appropriate to compare with our results, namely from the suggested authors (Szczebyło et al., 2020; Śmiglak-Krajewska et al, 2020, Jerzak et al., 2020, Melendrez-Ruiz et al., 2020; Figueira et al., 2019) WE have now included them in our discussion. Moreover, we tried to contextualize with national reality regarding culture and national policies. Finally, we rised new research questions that can conduct to an effective increase in pulses’ consumption.
From the discussion, the LIMITATION sections should be separated as a separate item.
A: Alterations have been performed according to the suggestion.

Reviewer 2 Report
Thank you for the opportunity to review this manuscript detailing pulse consumption among Portuguese adults. Although this manuscript describes an important topic of driver and barriers towards increased pulse consumption, the manuscript lacks detailed methodology. This major limitation makes it challenging to fully evaluate the results and discussion.
Line 41 – Can you provide a numeric description of how deficient the Portuguese intake of pulses is. Similar to the numeric description for the “meat,fish,eggs” group.
Line 43 – Are there any data to show how the intake of pulses has decreased over time as individuals moved away from ancestral diets?
Methods and Materials – This section is quite brief and lacking in critical details.
Line 74 – Information regarding IRB approval/ ethics committee?
Line 77 – Please provide a reference to the snowball sampling method.
Line 77 – Can the authors provide additional information about how the questionnaire was distributed? Saying only “disclosed through email and Facebook” is quite vague.
Line 78 – How was inclusion criteria confirmed, especially considering the method of questionnaire distribution?
Line 80 – Please provide additional details about the questionnaire. How many questions were there? How many questions asked about pulse freq and readiness for inclusion? How were these questions phrased?
Line 80 – Were these questionnaires validated? What is the reproducibility of responses?
Line 80 – The authors may want to consider adding parts of their questionnaire as supplemental figures.
Line 81 – How was the questionnaire delivered to participants? Was a particular software utilized? These details must be included.
Line 82 – Can the authors comment to some extent of how many people the questionnaire was distributed to?
Line 83 – Recommend creating subsections to divide quantitative and qualitative methods
Line 85 – what does professional intervention areas mean?
Line 91 – What types of questions were used to collect behaviors, attitudes, and experiences?
Line 92 – Please provide in a table these six open-ended questions.
Line 92 – How were these questions developed?
Line 93 – Who recorded and transcribed?
Line 95 – Recommend creating a subsection for statistical analysis
Line 97 – Please elaborate about how this method was specifically used with the qualitative data.
Table 2 – Please replace all “,” with decimals within percentages.
Figure 1 – Please make the graphic icons in the figure legend larger. Currently they are much too small to be decipherable. Consider using color even as the publisher does not charge for this.
Line 112 – The phrasing of the question requires a high reading level using words like “hypothesis”. Usually questionnaire items are written at a lower reading level. This may be limited understanding of questionnaire items and should be addressed as a limitation.
Given that the article includes qualitative data, the authors may want to use a research checklist such as the COREQ (COnsolidated criteria for REporting Qualitative research) checklist to ensure that all critical details are included.
Overall, the discussion should be restructured to describe the study’s findings in context with the existing literature in the order that it’s presented in the results. The quantitative data should be discussed first followed by the qualitative data.
In the discussion it should be mentioned how the Portuguese diet has evolved over time. It used to include many pulses, but has since strayed. Why was this? Could this be useful information to help the population return to traditional eating patterns?
Line 310 – Describe these strengths of the study.
Line 319 – What about a limitation of respondents not being able to adequately describe their usual intake of pulses?
Line 319 - Since the sample is quite young and largely female, can the authors speculate how this might reflect on the population as a whole. Would results be the same if the population was older? More men included? Broader geographic sample? Are there any similar studies to reflect opinions within these groups to substantiate the current findings?
Line 325 – Based on the information gleaned from this study what should the focus of these education sessions be? Should the focus be on the health benefits? Sustainability benefits? Some other benefits?
Line 329-Line 333 This is quite vague. Can the authors elaborate on more specific agricultural/nutritional recommendations?
Author Response
Response to reviewer comments
We thank all the valuable comments and the opportunity to improve our manuscript. Please, find below our answer to your comments and the respective indications about the changes operated in the manuscript. We have put the reviewers’ comments with italic font and our answer start with “A:”. In attach you will find the revised version of the manuscript.
Reviewer #2: Comments for authors
Thank you for the opportunity to review this manuscript detailing pulse consumption among Portuguese adults. Although this manuscript describes an important topic of driver and barriers towards increased pulse consumption, the manuscript lacks detailed methodology. This major limitation makes it challenging to fully evaluate the results and discussion.
Line 41 – Can you provide a numeric description of how deficient the Portuguese intake of pulses is. Similar to the numeric description for the “meat, fish,eggs” group.
A: Thanks for your observation. We completed now this information adding: “…, accounting with only 0.6% of the total amount of recommended foods, when the recommendation is 4%” (lines 41, 42).
Line 43 – Are there any data to show how the intake of pulses has decreased over time as individuals moved away from ancestral diets?
A: In Portugal we have a deficit of data about food consumption. Until now, we realized only two national food surveys: one in 1980 and another in 2015-2016. However, considering food sheets as a proxy of food intake, we observe that pulses consumption decreased significantly since 1961 until nowadays. This data was deeply analysed in a bachelor thesis, unfortunately written in Portuguese (available at: https://repositorio-aberto.up.pt/bitstream/10216/54664/3/127334_0933TCD33.pdf). We have now included a summary of these findings in the manuscript (lines 45-47)
Methods and Materials – This section is quite brief and lacking in critical details.
Line 74 – Information regarding IRB approval/ ethics committee?
A: As stated in lines 117 and 118: “Our study was approved by the Institute of Bioethics, Catholic University of Portugal, through the Ethics Screening Report 11/2017.” Regarding IRB, in our Faculty all master thesis are approved by our Scientific Council.
Line 77 – Please provide a reference to the snowball sampling method.
A: We have now include a reference for the methodology in line 84.
Line 77 – Can the authors provide additional information about how the questionnaire was distributed? Saying only “disclosed through email and Facebook” is quite vague.
A: We have added additional information. In lines 84-87, we added: “Questionnaires were firstly disclosed amongst the Faculty community (~800 individuals, including students, academic and administrative staff, and researchers) and subsequently among social networks of principal researchers. This contact had two objectives: to request the participation in the study and to request it’s widespread dissemination amongst participants’ contacts.”
Line 78 – How was inclusion criteria confirmed, especially considering the method of questionnaire distribution?
A: The inclusion criteria was for participants to be aged 18 or over and residents in Portugal. The introduction of the questionnaire mentioned these inclusion criteria. Later, in the questions about sociodemographic data, the respondent's age and geographic location was asked. In case of not meeting the inclusion criteria, questionnaires were eliminated from the database. This information has been added to the materials and methods section (88-91).
Line 80 – Please provide additional details about the questionnaire. How many questions were there? How many questions asked about pulse freq and readiness for inclusion? How were these questions phrased?
A: The complete questionnaire was composed of 24 questions, and it aimed to assess acceptability of legumes and other alternatives to animal protein, namely algae and insects (the latter two were not analysed or included in the current manuscript). For the purpose of the current study, the questionnaire included four socioeconomic questions (age, sex, education and local of residence); one question about self-perceived food pattern; one table to assess pulses’ frequency intake – with the six items presented in table 2, but considering seven options for frequency of consumption (from “never” to “more than once a day”); and a question about the readiness to include pulses as a partial substitute of animal protein (a closed question with five options, presented in figure 1). This information was added in the manuscript, lines 93-97.
Line 80 – Were these questionnaires validated? What is the reproducibility of responses?
A: The questionnaire for the quantitative study was constructed by the research team and appreciated by three specialists in the field of human nutrition and plant nutrition. Their inputs were incorporated. Subsequently, a pilot study was conducted among the students of the Epidemiology Curricular Unit (n=40). New inputs arose and were incorporated in the final version. Due to time constrains, reproducibility was not assessed, and we considered very likely that the respondents would recall the previous answers, since we collected the data in two weeks. Questions used in the qualitative study were also appreciated by an expert in social sciences. All authors read the interviews transcribed and contributed to the analysis.
Line 80 – The authors may want to consider adding parts of their questionnaire as supplemental figures.
A: We consider that after adding the detailed descriptions and corrections suggested by the reviewers, the manuscript methodology and the questionnaire itself are now better described in the manuscript and thus do not warrant an additional figure.
Line 81 – How was the questionnaire delivered to participants? Was a particular software utilized? These details must be included.
A: We created the questionnaire in Google Forms and the respective link was widespread. We added this information in the manuscript (lines 97-98): “Questionnaire was prepared in Google forms and respective link was widespread.”
Line 82 – Can the authors comment to some extent of how many people the questionnaire was distributed to?
A: Due to the strategy used to distribute the questionnaire, unfortunately it is not possible to check how many people received the questionnaire and thus how many people chose not to answer. This limitation is associated with the use of a social network, namely when the snowball method is incited.
Line 83 – Recommend creating subsections to divide quantitative and qualitative methods
A: We thank your suggestion and we have operated this division in the present version.
Line 85 – what does professional intervention areas mean?
R: For the qualitative study, we selected individuals dealing with pulses at different steps of the value chain, from production to the consumption. First, we identified the stage that we want to be represented and, secondly, we identified a person working in this specific stage. This has been better described in 103-106.
Line 91 – What types of questions were used to collect behaviors, attitudes, and experiences? Line 92 – Please provide in a table these six open-ended questions.
A: Thanks for your suggestion. We added a table with this information in the manuscript (table 1). In the table reviewers could see the type of questions used to collect behaviors, attitudes and experiences.
Line 92 – How were these questions developed?
A: The questions were developed based on quantitative data from the Legume Innovation Workshops of the TRUE project, since we wanted to ascertain what pulses represented for the participants and, since they had experience in the field, to know their suggestions to increase pulses’ consumption. We started to define the domains and, after, the sub-domains, as presented in table 1.
Line 93 – Who recorded and transcribed?
A: Interviews were recorded and transcribed by the principal researcher, who also conducted the interviews. This information is in line 113-114.
Line 95 – Recommend creating a subsection for statistical analysis
R: Thank you for the suggestion. We have included it in the present version.
Line 97 – Please elaborate about how this method was specifically used with the qualitative data.
R: Content analysis included research techniques that allowed, in a systematic way, a description of the messages and attitudes related to the context. This method comprises a set of procedures for organizing information, using the collected qualitative data to create themes or categories in order to sum them up. In detail, we started with a pre-analysis that consisted in a free-floating reading and in the elaboration of the analysis structure (categories and sub-categories), that we had defined previously. After, we explored the interviews and codified the pieces of text that could be allocated to each sub-category. Finally, we interpreted data merging theoretical concepts and empirical data. We added this procedure to the manuscript (lines 125-130).
Table 2 – Please replace all “,” with decimals within percentages.
A: Thank you for the observation. We have altered according to the suggestion.
Figure 1 – Please make the graphic icons in the figure legend larger. Currently they are much too small to be decipherable. Consider using color even as the publisher does not charge for this.
A: The alteration has been performed according to the suggestion.
Line 112 – The phrasing of the question requires a high reading level using words like “hypothesis”. Usually questionnaire items are written at a lower reading level. This may be limited understanding of questionnaire items and should be addressed as a limitation.
A: We understand the reviewer´s concern. In fact, “hypothesis” is not the best translation for the term used in the questionnaire, which was in Portuguese. In Portuguese we actually asked if participants would “consider the possibility”. This has been corrected in the manuscript. In the Portuguese language, “put the hypothesis” is a simple synonym of “to consider the possibility” and this expression in common in current speech.
Given that the article includes qualitative data, the authors may want to use a research checklist such as the COREQ (COnsolidated criteria for Reporting Qualitative research) checklist to ensure that all critical details are included.
A: Unfortunately we did not consider a priori an instrument like COREQ when we conducted our analysis, but after your comment we found a COREQ checklist (Tong A et al. Consolidated criteria for reporting qualitative research (COREQ): a 32-item checklist for interviews and focus groups. International Journal for Quality in Health Care 2007; 19(6): 349-57) and we can confirm that we realised all the proposed steps. We have added this information in the methods (lines 115-116).
Overall, the discussion should be restructured to describe the study’s findings in context with the existing literature in the order that it’s presented in the results. The quantitative data should be discussed first followed by the qualitative data.
A: We agree with your comment. We restructured the order in which we present the discussion and we added additional references to compare with our data and we separated the discussion of qualitative and quantitative data, as suggested. Changes are marked with track changes in the text.
In the discussion it should be mentioned how the Portuguese diet has evolved over time. It used to include many pulses, but has since strayed. Why was this? Could this be useful information to help the population return to traditional eating patterns?
A: We agree that this information is important in the discussion. We included it in lines 273-281.
Line 310 – Describe these strengths of the study.
A: The present study helps to fill gaps in scientific literature on the motivations that lead to the current drop in the pulses consumption. It provides answers to questions regarding prevalence of willingness for the replacement of meat or fish by pulses and data about barriers to the consumption of this food in Portuguese individuals. This useful information helps international and local companies to develop and implement new ways of looking at pulses and reinventing forms that suit the lifestyle followed by the Portuguese consumers. Moreover, this study generated new hypotheses and prompt further, tailor-designed research. We added this information on lines 399-400.
Line 319 – What about a limitation of respondents not being able to adequately describe their usual intake of pulses?
A: We agree that this is a limitation and, now, it is included into limitations section (lines 373-375).
Line 319 - Since the sample is quite young and largely female, can the authors speculate how this might reflect on the population as a whole. Would results be the same if the population was older? More men included? Broader geographic sample? Are there any similar studies to reflect opinions within these groups to substantiate the current findings?
A: We can speculate that if we included in the sample older people, especially men, we would obtain similar frequencies of consumption, considering the results of the national food survey conducted in 2015 (Lopes et al.), but probably a higher reluctance to substitute animal sources of protein by pulses. This information was introduced in lines 395-398.
Line 325 – Based on the information gleaned from this study what should the focus of these education sessions be? Should the focus be on the health benefits? Sustainability benefits? Some other benefits?
R: After identifying the main barriers to the consumption of pulses, it was realized that it is probably necessary to create food education sessions focused on nutritional, health and sustainability benefits in order to demystify concepts rooted in the Portuguese population. However, we believe that it is also fundamental to provide recipes that are not only easy and quick to prepare, but also allow the inclusion of pulses on innovative and different forms of presentation from those already known and traditional. We re-phrase the second paragraph of conclusions to highlight these ideas.
Line 329-Line 333 This is quite vague. Can the authors elaborate on more specific agricultural/nutritional recommendations?
A: We completed the last paragraph of the conclusions section, in order to detail more about these recommendations.

Round 2
Reviewer 1 Report
The authors referred to the suggested corrections. A very well done job which improved the quality of the paper.
Reviewer 2 Report
Thank you to the authors for their point by point revision of the paper. I believe the additions to the manuscript improve the scientific soundness. I have no further comments.